# Non-Uniform Camera Shake Removal Using a Spatially-Adaptive Sparse Penalty

**Haichao Zhang**[†‡] **and David Wipf** [§]

[†] School of Computer Science, Northwestern Polytechnical University, Xi'an, China
[‡] Department of Electrical and Computer Engineering, Duke University, USA
[§] Visual Computing Group, Microsoft Research Asia, Beijing, China
hczhang1@gmail.com   davidwipf@gmail.com

## Abstract

Typical blur from camera shake often deviates from the standard uniform convolutional assumption, in part because of problematic rotations which create greater blurring away from some unknown center point. Consequently, successful blind deconvolution for removing shake artifacts requires the estimation of a spatially-varying or non-uniform blur operator. Using ideas from Bayesian inference and convex analysis, this paper derives a simple non-uniform blind deblurring algorithm with a spatially-adaptive image penalty. Through an implicit normalization process, this penalty automatically adjust its shape based on the estimated degree of local blur and image structure such that regions with large blur or few prominent edges are discounted. Remaining regions with modest blur and revealing edges therefore dominate on average without explicitly incorporating structure-selection heuristics. The algorithm can be implemented using an optimization strategy that is virtually tuning-parameter free and simpler than existing methods, and likely can be applied in other settings such as dictionary learning. Detailed theoretical analysis and empirical comparisons on real images serve as validation.

## 1   Introduction

Image blur is an undesirable degradation that often accompanies the image formation process and may arise, for example, because of camera shake during acquisition. Blind image deblurring strategies aim to recover a sharp image from only a blurry, compromised observation. Extensive efforts have been devoted to the uniform blur (shift-invariant) case, which can be described with the convolutional model $\mathbf{y} = \mathbf{k} * \mathbf{x} + \mathbf{n}$, where $\mathbf{x}$ is the unknown sharp image, $\mathbf{y}$ is the observed blurry image, $\mathbf{k}$ is the unknown blur kernel (or point spread function), and $\mathbf{n}$ is a zero-mean Gaussian noise term [6, 21, 17, 5, 28, 14, 1, 27, 29]. Unfortunately, many real-world photographs contain blur effects that vary across the image plane, such as when unknown rotations are introduced by camera shake [17].

More recently, algorithms have been generalized to explicitly handle some degree of non-uniform blur using the more general observation model $\mathbf{y} = \mathbf{Hx} + \mathbf{n}$, where each column of the blur operator $\mathbf{H}$ contains the spatially-varying effective blur kernel at the corresponding pixel site [25, 7, 8, 9, 11, 4, 22, 12]. Note that the original uniform blur model can be achieved equivalently when $\mathbf{H}$ is forced to adopt certain structure (e.g., block-toeplitz structure with toeplitz-blocks). In general, non-uniform blur may arise under several different contexts. This paper will focus on the blind removal of non-uniform blur caused by general camera shake (as opposed to blur from object motion) using only a single image, with no additional hardware assistance.

While existing algorithms for addressing non-uniform camera shake have displayed a measure of success, several important limitations remain. First, some methods require either additional spe-

cialized hardware such as high-speed video capture [23] or inertial measurement sensors [13] for estimating motion, or else multiple images of the same scene [4]. Secondly, even the algorithms that operate given only data from a single image typically rely on carefully engineered initializations, heuristics, and trade-off parameters for selecting salient image structure or edges, in part to avoid undesirable degenerate, no-blur solutions [7, 8, 9, 11]. Consequently, enhancements and rigorous analysis may be problematic. To address these shortcomings, we present an alternative blind deblurring algorithm built upon a simple, closed-form cost function that automatically discounts regions of the image that contain little information about the blur operator without introducing any additional salient structure selection steps. This transparency leads to a nearly tuning-parameter free algorithm based upon a sparsity penalty whose shape adapts to the estimated degree of local blur, and provides theoretical arguments regarding how to robustly handle non-uniform degradations.

The rest of the paper is structured as follows. Section 2 briefly describes relevant existing work on non-uniform blind deblurring operators and implementation techniques. Section 3 then introduces the proposed non-uniform blind deblurring model, while further theoretical justification and analyses are provided in Section 4. Experimental comparisons with *state-of-the-art* methods are carried out in Section 5 followed by conclusions in Section 6.

## 2   Non-Uniform Deblurring Operators

Perhaps the most direct way of handling non-uniform blur is to simply partition the image into different regions and then learn a separate, uniform blur kernel for each region, possibly with an additional weighting function for smoothing the boundaries between two adjacent kernels. The resulting algorithm has been adopted extensively [18, 8, 22, 12] and admits an efficient implementation called *efficient filter flow* (EFF) [10]. The downside with this type of model is that geometric relationships between the blur kernels of different regions derived from the the physical motion path of the camera are ignored.

In contrast, to explicitly account for camera motion, the *projective motion path* (PMP) model [23] treats a blurry image as the weighted summation of projectively transformed sharp images, leading to the revised observation model

$$\mathbf{y} = \sum_j w_j \mathbf{P}_j \mathbf{x} + \mathbf{n}, \tag{1}$$

where $\mathbf{P}_j$ is the $j$-th projection or homography operator (a combination of rotations and translations) and $w_j$ is the corresponding combination weight representing the proportion of time spent at that particular camera pose during exposure. The uniform convolutional model can be obtained by restricting the general projection operators $\{\mathbf{P}_j\}$ to be translations. In this regard, (1) represents a more general model that has been used in many recent non-uniform deblurring efforts [23, 25, 7, 11, 4]. PMP also retains the bilinear property of uniform convolution, meaning that

$$\mathbf{y} = \mathbf{H}\mathbf{x} + \mathbf{n} = \mathbf{D}\mathbf{w} + \mathbf{n}, \tag{2}$$

where $\mathbf{H} = \sum_j w_j \mathbf{P}_j$ and $\mathbf{D} = [\mathbf{P}_1\mathbf{x}, \mathbf{P}_2\mathbf{x}, \cdots, \mathbf{P}_j\mathbf{x}, \cdots]$ is a matrix of transformed sharp images.

The disadvantage of PMP is that it typically leads to inefficient algorithms because the evaluation of the matrix-vector product $\mathbf{H}\mathbf{x} = \mathbf{D}\mathbf{w}$ requires generating many expensive intermediate transformed images. However, EFF can be combined with the PMP model by introducing a set of basis images efficiently generated by transforming a grid of delta peak images [9]. The computational cost can be further reduced by using an active set for pruning out the projection operators with small responses [11].

## 3   A New Non-Uniform Blind Deblurring Model

Following previous work [6, 16], we will work in the derivative domain of images for ease of modeling and better performance, meaning that $\mathbf{x} \in \mathbb{R}^m$ and $\mathbf{y} \in \mathbb{R}^n$ will denote the lexicographically ordered sharp and blurry image derivatives respectively. [1]

The observation model (1) is equivalent to the likelihood function

$$p(\mathbf{y}|\mathbf{x},\mathbf{w}) \propto \exp\left[-\frac{1}{2\lambda}\|\mathbf{y}-\mathbf{Hx}\|_2^2\right], \tag{3}$$

where $\lambda$ denotes the noise variance. Maximum likelihood estimation of $\mathbf{x}$ and $\mathbf{w}$ using (3) is clearly ill-posed and so further regularization is required to constrain the solution space. For this purpose we adopt the Gaussian prior $p(\mathbf{x}) \sim \mathcal{N}(\mathbf{x};\mathbf{0},\mathbf{\Gamma})$, where $\mathbf{\Gamma} \triangleq \mathrm{diag}[\boldsymbol{\gamma}]$ with $\boldsymbol{\gamma} = [\gamma_1,\ldots,\gamma_m]^T$ a vector of $m$ hyperparameter variances, one for each element of $\mathbf{x} = [x_1,\ldots,x_m]^T$. While presently $\boldsymbol{\gamma}$ is unknown, if we first marginalize over the unknown $\mathbf{x}$, we can estimate it jointly along with the blur parameters $\mathbf{w}$ and the unknown noise variance $\lambda$. This *type II maximum likelihood* procedure has been advocated in the context of sparse estimation, where the goal is to learn vectors with mostly zero-valued coefficients [24, 26]. The final sharp image can then be recovered using the estimated kernel and noise level along with standard non-blind deblurring algorithms (e.g., [15]).

Mathematically, the proposed estimation scheme requires that we solve

$$\max_{\boldsymbol{\gamma},\mathbf{w},\lambda\geq 0} \int p(\mathbf{y}|\mathbf{x},\mathbf{w})p(\mathbf{x})d\mathbf{x} \equiv \min_{\boldsymbol{\gamma},\mathbf{w},\lambda\geq 0} \mathbf{y}^T\left(\mathbf{H\Gamma H}^T+\lambda\mathbf{I}\right)^{-1}\mathbf{y} + \log\left|\mathbf{H\Gamma H}^T+\lambda\mathbf{I}\right|, \tag{4}$$

where a $-\log$ transformation has been included for convenience. Clearly (4) does not resemble the traditional blind non-uniform deblurring script, where estimation proceeds using the more transparent penalized regression model [4, 7, 9]

$$\min_{\mathbf{x};\mathbf{w}\geq 0} \|\mathbf{y}-\mathbf{Hx}\|_2^2 + \alpha\sum_i g(x_i) + \beta\sum_j h(w_j) \tag{5}$$

and $\alpha$ and $\beta$ are user-defined trade-off parameters, $g$ is an image penalty which typically favors sparsity, and $h$ is usually assumed to be quadratic. Despite the differing appearances however, (4) has some advantageous properties with respect to deconvolution problems. In particular, it is devoid of tuning parameters and it possesses more favorable minimization conditions. For example, consider the simplified non-uniform deblurring situation where the true $\mathbf{x}$ has a single non-zero element and $\mathbf{H}$ is defined such that each column indexed by $i$ is independently parameterized with finite support symmetric around pixel $i$. Moreover, assume this support matches the true support of the unknown blur operator. Then we have the following:

**Lemma 1** *Given the idealized non-uniform deblurring problem described above, the cost function (4) will be characterized by a unique minimizing solution that correctly locates the nonzero element in $\mathbf{x}$ and the corresponding true blur kernel at this location. No possible problem in the form of (5), with $g(x) = |x|^p$, $h(w) = w^q$, and $\{p,q\}$ arbitrary non-negative scalars, can achieve a similar result (there will always exist either multiple different minimizing solutions or an global minima that does not produce the correct solution).*

This result, which can be generalized with additional effort, can be shown by expanding on some of the derivations in [26]. Although obviously the conditions upon which Lemma 1 is based are extremely idealized, it is nonetheless emblematic of the potential of the underlying cost function to avoid local minima, etc., and [26] contains complementary results in the case where $\mathbf{H}$ is fixed.

While optimizing (4) is possible using various general techniques such as the EM algorithm, it is computationally expensive in part because of the high-dimensional determinants involved with realistic-sized images. Consequently we are presently considering various specially-tailored optimization schemes for future work. But for the present purposes, we instead minimize a convenient upper bound allowing us to circumvent such computational issues. Specifically, using Hadamard's inequality we have

$$\begin{aligned}
\log\left|\mathbf{H\Gamma H}^T+\lambda\mathbf{I}\right| &= n\log\lambda + \log|\mathbf{\Gamma}| + \log\left|\lambda^{-1}\mathbf{H}^T\mathbf{H}+\mathbf{\Gamma}^{-1}\right| \\
&\leq n\log\lambda + \log|\mathbf{\Gamma}| + \log\left|\lambda^{-1}\mathrm{diag}\left[\mathbf{H}^T\mathbf{H}\right]+\mathbf{\Gamma}^{-1}\right| \\
&= \sum_i \log\left(\lambda+\gamma_i\|\bar{\mathbf{w}}_i\|_2^2\right) + (n-m)\log\lambda, \tag{6}
\end{aligned}$$

where $\bar{\mathbf{w}}_i$ denotes the $i$-th column of $\mathbf{H}$. Note that Hadamard's inequality is applied by using $\lambda^{-1}\mathbf{H}^T\mathbf{H}+\mathbf{\Gamma}^{-1} = \mathbf{V}^T\mathbf{V}$ for some matrix $\mathbf{V} = [\mathbf{v}_1,\ldots,\mathbf{v}_m]$. We then have $\log|\lambda^{-1}\mathbf{H}^T\mathbf{H}+\mathbf{\Gamma}^{-1}| = 2\log|\mathbf{V}| \leq 2\log\left(\prod_i\|\mathbf{v}_i\|_2\right) = \log\left|\mathrm{diag}\left[\lambda^{-1}\mathbf{H}^T\mathbf{H}+\mathbf{\Gamma}^{-1}\right]\right|$, leading to the stated result.

Also, the quantity $\|\bar{\mathbf{w}}_i\|_2$ which appears in (6) can be viewed as a measure of the degree of local blur at location $i$. Given the feasible region $\mathbf{w} \geq 0$ and without loss of generality the constraint $\sum_i w_i = 1$ for normalization purposes, it can easily be shown that $1/L \leq \|\bar{\mathbf{w}}_i\|_2^2 \leq 1$, where $L$ is the maximum number of elements in any local blur kernel $\bar{\mathbf{w}}_i$ or column of $\mathbf{H}$. The upper bound is achieved when the local kernel is a delta solution, meaning only one nonzero element and therefore minimal blur. In contrast, the lower bound on $\|\bar{\mathbf{w}}_i\|_2^2$ occurs when every element of $\bar{\mathbf{w}}_i$ has an equal value, constituting the maximal possible blur. This metric, which will influence our analysis in the next section, can be computing using $\|\bar{\mathbf{w}}_i\|_2^2 = \mathbf{w}^T(\mathbf{B}_i^T\mathbf{B}_i)\mathbf{w}$, where $\mathbf{B}_i \triangleq [\mathbf{P}_1\mathbf{e}_i, \mathbf{P}_2\mathbf{e}_i, \cdots, \mathbf{P}_j\mathbf{e}_i, \cdots]$ and $\mathbf{e}_i$ denotes an all-zero image with a one at site $i$. In the uniform deblurring case, $\mathbf{B}_i^T\mathbf{B}_i = \mathbf{I}$ ignoring edge effects, and therefore $\|\bar{\mathbf{w}}_i\|_2 = \|\mathbf{w}\|_2$ for all $i$.

While optimizing (4) using the upper bound from (6) can be justified in part using Bayesian-inspired arguments and the lack of trade-off parameters, the augmented cost function unfortunately no longer satisfies Lemma 1. However, it is still well-equipped for estimating sparse image gradients and avoiding degenerate no-blur solutions. For example, consider the case of an asymptotically large image with iid distributed sparse image gradients, with some constant fraction exactly equal to zero and the remaining nonzero elements drawn from any continuous distribution. Now suppose that this image is corrupted with a non-uniform blur operator of the form $\mathbf{H} = \sum_j w_j\mathbf{P}_j$, where the cardinality of the summation is finite and $\mathbf{H}$ satisfies minimal regularity conditions. Then it can be shown that any global minimum of (4), with or without the bound from (6), will produce the true blur operator. Related intuition applies when noise is present or when the image gradients are not exactly sparse (we will defer more detailed analysis to a future publication).

Regardless, the simplified $\gamma$-dependent cost function is still far less intuitive than the penalized regression models dependent on $\mathbf{x}$ such as (5) that are typically employed for non-uniform blind deblurring. However, using the framework from [26], it can be shown that the kernel estimate obtained by this process is formally equivalent to the one obtained via

$$\min_{\mathbf{x};\mathbf{w}\geq 0,\lambda\geq 0} \frac{1}{\lambda}\|\mathbf{y} - \mathbf{Hx}\|_2^2 + \sum_i \psi(|x_i|\|\bar{\mathbf{w}}_i\|_2, \lambda) + (n-m)\log\lambda, \qquad \text{with} \qquad (7)$$

$$\psi(u, \lambda) \triangleq \frac{2u}{u + \sqrt{4\lambda + u^2}} + \log\left(2\lambda + u^2 + u\sqrt{4\lambda + u^2}\right) \quad u \geq 0.$$

The optimization from (7) closely resembles a standard penalized regression (or equivalently MAP) problem used for blind deblurring. The primary distinction is the penalty term $\psi$, which jointly regularizes $\mathbf{x}$, $\mathbf{w}$, and $\lambda$ as discussed Section 4. The supplementary file derives a simple majorization-minimization algorithm for solving (7) along with additional implementational details. The underlying procedure is related to variational Bayesian (VB) models from [1, 16, 20]; however, these models are based on a completely different mean-field approximation and a uniform blur assumption, and they do not learn the noise parameter. Additionally, the analysis provided with these VB models is limited by relatively less transparent underlying cost functions.

## 4  Model Properties

The proposed blind deblurring strategy involves simply minimizing (7); no additional steps for trade-off parameter selection or structure/salient-edge detection are required unlike other state-of-the-art approaches. This section will examine theoretical properties of (7) that ultimately allow such a simple algorithm to succeed. First, we will demonstrate a form of intrinsic column normalization that facilitates the balanced sparse estimation of the unknown latent image and implicitly de-emphasizes regions with large blur and few dominate edges. Later we describe an appealing form of noise-dependent shape adaptation that helps in avoiding local minima. While there are multiple, complementary perspectives for interpreting the behavior of this algorithm, more detailed analyses, as well as extensions to other types of underdetermined inverse problems such as dictionary learning, will be deferred to a later publication.

### 4.1  Column-Normalized Sparse Estimation

Using the simple reparameterization $z_i \triangleq x_i\|\bar{\mathbf{w}}_i\|_2$ it follows that (7) is exactly equivalent to solving

$$\min_{\mathbf{z};\mathbf{w}\geq 0,\lambda\geq 0} \frac{1}{\lambda}\|\mathbf{y} - \widetilde{\mathbf{H}}\mathbf{z}\|_2^2 + \sum_i \psi(|z_i|, \lambda) + (n-m)\log\lambda, \qquad (8)$$

where $\mathbf{z} = [z_1, \ldots, z_m]^T$ and $\widetilde{\mathbf{H}}$ is simply the $\ell_2$-column-normalized version of $\mathbf{H}$. Moreover, it can be shown that this $\psi$ is a concave, non-decreasing function of $|z|$, and hence represents a canonical sparsity-promoting penalty function with respect to $\mathbf{z}$ [26]. Consequently, noise and kernel dependencies notwithstanding, this reparameterization places the proposed cost function in a form exactly consistent with nearly all prototypical sparse regression problems, where $\ell_2$ column normalization is ubiquitous, at least in part, to avoid favoring one column over another during the estimation process (which can potentially bias the solution). To understand the latter point, note that $\|\mathbf{y} - \widetilde{\mathbf{H}}\mathbf{z}\|_2^2 \equiv \mathbf{z}^T\widetilde{\mathbf{H}}^T\widetilde{\mathbf{H}}\mathbf{z} - 2\mathbf{y}^T\widetilde{\mathbf{H}}\mathbf{z}$. Among other things, because of the normalization, the quadratic factor $\widetilde{\mathbf{H}}^T\widetilde{\mathbf{H}}$ now has a unit diagonal, and likewise the inner products $\mathbf{y}^T\widetilde{\mathbf{H}}$ are scaled by the consistent induced $\ell_2$ norms, which collectively avoids the premature favoring of any one element of $\mathbf{z}$ over another. Moreover, no additional heuristic kernel penalty terms such as in (5) are required since $\widetilde{\mathbf{H}}$ is in some sense self-regularized by the normalization. Additional ancillary benefits of (8) will be described in Section 4.2.

Of course we can always apply the same reparameterization to existing algorithms in the form of (5). While this will indeed result in normalized columns and a properly balanced data-fit term, these raw norms will now appear in the penalty function $g$, giving the equivalent objective

$$\min_{\mathbf{z};\mathbf{w}\geq 0} \|\mathbf{y} - \widetilde{\mathbf{H}}\mathbf{z}\|_2^2 + \alpha \sum_i g\left(z_i\|\bar{\mathbf{w}}_i\|_2^{-1}\right) + \beta \sum_j h(w_j). \tag{9}$$

However, the presence of these norms now embedded in $g$ may have undesirable consequences. Simply put, the problem (9) will favor solutions where the ratio $z_i/\|\bar{\mathbf{w}}_i\|_2$ is sparse or nearly so, which can be achieved by either making many $z_i$ zero or many $\|\bar{\mathbf{w}}_i\|_2$ big. If some $z_i$ is estimated to be zero (and many $z_i$ will provably be exactly zero at any local minima if $g(x)$ is a concave, non-decreasing function of $|x|$), then the corresponding $\|\bar{\mathbf{w}}_i\|_2$ will be unconstrained. In contrast, if a given $z_i$ is non-zero, there will be a stronger push for the associated $\|\bar{\mathbf{w}}_i\|_2$ to be large, i.e., more like the delta kernel which maximizes the $\ell_2$ norm. Thus, the relative penalization of the kernel norms will depend on the estimated local image gradients, and no-blur delta solutions may be arbitrarily favored in parts of the image plane dominated by edges, the very place where blur estimation information is paramount.

In reality, the local kernel norms $\|\bar{\mathbf{w}}_i\|_2$, which quantify the degree of local blur as mentioned previously, should be completely independent of the sparsity of the image gradients in the same location. This is of course because the different blurring effects from camera shake are independent of the locations of strong edges in a given scene, since the blur operator is only a function of camera motion (at least to first order approximation). One way to compensate for this independence would be to simply optimize (9) with $\|\bar{\mathbf{w}}_i\|_2$ removed from $g$. While this is possible in principle, enforcing the non-convex, and coupled constraints required to maintain normalized columns is extremely difficult. Another option would be to carefully choose $\beta$ and $h$ to somehow compensate. In contrast, our algorithm handles these complications seamlessly without any additional penalty terms.

## 4.2 Noise-Dependent, Parameter-Free Homotopy Continuation

Column normalization can be viewed as a principled first step towards solving challenging sparse estimation problems. However, when non-convex sparse regularizers are used for the image penalty, e.g., $\ell_p$ norms with $p < 1$, then local minima can be a significant problem. The rationalization for using such potentially problematic non-convexity is as follows; more details can be found in [17, 27]. When applied to a sharp image, any blur operator will necessarily contribute two opposing effects: (i) It reduces a measure of the image *sparsity*, which normally increases the penalty $\sum_i |y_i|^p$, and (ii) It broadly reduces the overall image *variance*, which actually reduces $\sum_i |y_i|^p$. Additionally, the greater the degree of blur, the more effect (ii) will begin to overshadow (i). Note that we can always apply greater and greater blur to any sharp image $\mathbf{x}$ such that the variance of the resulting blurry $\mathbf{y}$ is arbitrarily small. This then produces an arbitrarily small $\ell_p$ norm, which implies that $\sum_i |y_i|^p < \sum_i |x_i|^p$, meaning that the penalty actually favors the blurry image over the sharp one.

In a practical sense though, the amount of blur that can be tolerated before this undesirable preference for $\mathbf{y}$ over $\mathbf{x}$ occurs is much larger as $p$ approaches zero. This is because the more concave the image penalty becomes (as a function of coefficient magnitudes), the less sensitive it is to image variance and the more sensitive it is to image sparsity. In fact the scale-invariant special case where

$p \rightarrow 0$ depends only on *sparsity*, or the number of elements that are exactly equal to zero. [2] We may therefore expect such a highly concave, sparsity promoting penalty to favor the sharp image over the blurry one in a broader range of blur conditions. Even with other families of penalty functions the same basic notion holds: greater concavity means greater sparsity preference and less sensitivity to variance changes that favor no-blur degenerate solutions.

From an implementational standpoint, *homotopy continuation* methods provide one attractive means of dealing with difficult non-convex penalty functions and the associated constellation of local minima [3]. The basic idea is to use a parameterized family of sparsity-promoting functions $g(\mathbf{x}; \theta)$, where different values of $\theta$ determine the relative degree of concavity allowing a transition from something convex such as the $\ell_1$ norm (with $\theta$ large) to something concave such as the $\ell_0$ norm (with $\theta$ small). Moreover, to ensure cost function descent (see below), we also require that $g(\mathbf{x}; \theta_2) \geq g(\mathbf{x}; \theta_1)$ whenever $\theta_2 \geq \theta_1$, noting that this rules out simply setting $\theta = p$ and using the family of $\ell_p$ norms. We then begin optimization with a large $\theta$ value; later as the estimation progresses and hopefully we are near a reasonably good basin of attraction, $\theta$ is reduced introducing greater concavity, a process which is repeated until convergence, all the while guaranteeing cost function descent. While potentially effective in practice, homotopy continuation methods require both a trade-off parameter for $g(\mathbf{x}; \theta)$ and a pre-defined schedule or heuristic for adjusting $\theta$, both of which could potentially be image dependent.

The proposed deblurring algorithm automatically implements a form of noise-dependent, parameter-free homotopy continuation with several attractive auxiliary properties [26]. To make this claim precise and facilitate subsequent analysis, we first introduce the definition of *relative concavity* [19]:

**Definition 1** *Let $u$ be a strictly increasing function on $[a, b]$. The function $\nu$ is **concave relative** to $u$ on the interval $[a, b]$ if and only if $\nu(y) \leq \nu(x) + \frac{\nu'(x)}{u'(x)} [u(y) - u(x)]$ holds $\forall x, y \in [a, b]$.*

We will use $\nu \prec u$ to denote that $\nu$ is concave relative to $u$ on $[0, \infty)$. This can be understood as a natural generalization of the traditional notion of a concavity, in that a concave function is equivalently *concave relative to a linear function* per Definition 1. In general, if $\nu \prec u$, then when $\nu$ and $u$ are set to have the same functional value and the same slope at any given point (i.e., by an affine transformation of $u$), then $\nu$ lies completely under $u$. In the context of homotopy continuation, an ideal candidate penalty would be one for which $g(\mathbf{x}; \theta_1) \prec g(\mathbf{x}; \theta_2)$ whenever $\theta_1 \leq \theta_2$. This would ensure that greater sparsity-inducing concavity is introduced as $\theta$ is reduced. We now demonstrate that $\psi(|z|, \lambda)$ is such a function, with $\lambda$ occupying the role of $\theta$. This dependency on the noise parameter is unlike other continuation methods and ultimately leads to several attractive attributes.

**Theorem 1** *If $\lambda_1 < \lambda_2$, then $\psi(u, \lambda_1) \prec \psi(u, \lambda_2)$ for $u \geq 0$. Additionally, in the limit as $\lambda \rightarrow 0$, then $\sum_i \psi(|z_i|, \lambda)$ converges to the $\ell_0$ norm (up to an inconsequential scaling and translation). Conversely, as $\lambda$ becomes large, $\sum_i \psi(|z_i|, \lambda)$ converges to $2\|\mathbf{z}\|_1 / \sqrt{\lambda}$.*

The proof has been deferred to the supplementary file. The relevance of this result can be understood as follows. First, at the beginning of the optimization process $\lambda$ will be large both because of initialization and because we have not yet found a relatively sparse $\mathbf{z}$ and associated $\mathbf{w}$ such that $\mathbf{y}$ can be well-approximated; hence the estimated $\lambda$ should not be small. Based on Theorem 1, in this regime (8) approaches

$$\min_{\mathbf{z}} \|\mathbf{y} - \widetilde{\mathbf{H}}\mathbf{z}\|_2^2 + 2\sqrt{\lambda}\|\mathbf{z}\|_1 \qquad (10)$$

assuming $\mathbf{w}$ and $\lambda$ are fixed. Note incidentally that this square-root dependency on $\lambda$, which arises naturally from our model, is frequently advocated when performing regular $\ell_1$-norm penalized sparse regression given that the true noise variance is $\lambda$ [2]. Additionally, because $\lambda$ must be relatively large to arrive at this $\ell_1$ approximation, the estimation need only focus on reproducing the largest elements in $\mathbf{z}$ since the sparse penalty will dominate the data fit term. Furthermore, these larger elements are on average more likely to be in regions of relatively lower blurring or high $\|\bar{\mathbf{w}}_i\|_2$ value by virtue of the reparameterization $z_i = x_i \|\bar{\mathbf{w}}_i\|_2$. Consequently, the less concave initial estimation can proceed successfully by de-emphasizing regions with high blur or low $\|\bar{\mathbf{w}}_i\|_2$, and focusing on coarsely approximating regions with relatively less blur.

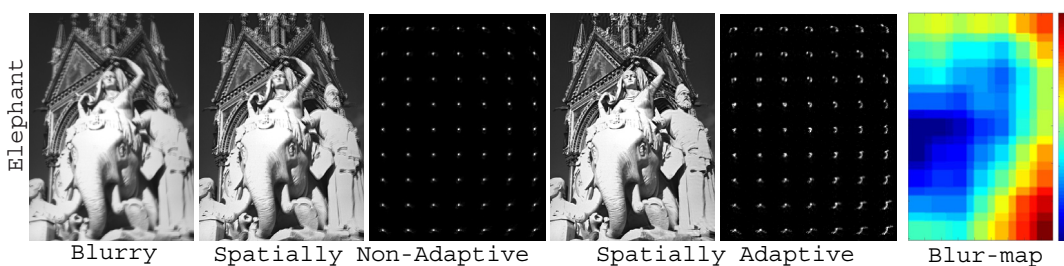

Figure 1: **Effectiveness of spatially-adaptive sparsity.** *From left to right*: the blurry image, the deblurred image and estimated local kernels without spatially-adaptive column normalization, the analogous results with this normalization and its spatially-varying impact on image estimation, and the associated map of $\|\bar{\mathbf{w}}_i\|_2^{-1}$, which reflects the degree of estimated local blurring.

Later as the estimation proceeds and $\mathbf{w}$ and $\mathbf{z}$ are refined, $\lambda$ will be reduced which in turn necessarily increases the relative concavity of the penalty $\psi$ per Theorem 1. However, the added concavity will now be welcome for resolving increasingly fine details uncovered by a lower noise variance and the concomitant boosted importance of the data fidelity term, especially since many of these uncovered details may reside near increasingly blurry regions of the image and we need to avoid unwanted no-blur solutions. Eventually the penalty can even approach the $\ell_0$ norm (although images are generally not exactly sparse, and other noise factors and unmodeled artifacts are usually present such that $\lambda$ will never go all the way to zero). Importantly, all of this implicit, spatially-adaptive penalization occurs without the need for trade-off parameters or additional structure selection measures, meaning carefully engineered heuristics designed to locate prominent edges such that good global solutions can be found without strongly concave image penalties [21, 5, 28, 8, 9]. Figure 1 displays results of this procedure both with and without the spatially-varying column normalizations and the implicit adaptive penalization that help compensate for locally varying image blur.

## 5 Experimental Results

This section compares the proposed method with several *state-of-the-art* algorithms for non-uniform blind deblurring using real-world images from previously published papers (note that source code is not available for conducting more widespread evaluations with most algorithms). The supplementary file contains a number of additional comparisons, including assessments with a benchmark uniform blind deblurring dataset where ground truth is available. Overall, our algorithm consistently performs comparably or better on all of these respective images. Experimental specifics of our implementation (e.g., regarding the non-blind deblurring step, projection operators, etc.) are also contained in the supplementary file for space considerations.

**Comparison with Harmeling *et al.* [8] and Hirsch *et al.* [9]:** Results are based on three test images provided in [8]. Figure 2 displays deblurring comparisons based on the `Butchershop` and `Vintage-car` images. In both cases, the proposed algorithm reveals more fine details than the other methods, despite its simplicity and lack of salient structure selection heuristics or trade-off parameters. Note that with these images, ground truth blur kernels were independently estimated using a special capturing process [8]. As shown in the supplementary file, the estimated blur kernel patterns obtained from our algorithm better resemble the ground truth relative to the other methods, a performance result that compensates for any differences in the non-blind step.

**Comparison with Whyte *et al.* [25]:** Results on the `Pantheon` test image from [25] are shown in Figure 3 (*top row*), where we observe that the deblurred image from Whyte *et al.* has noticeable ringing artifacts. In contrast, our result is considerably cleaner.

**Comparison with Gupta *et al.* [7]:** We next experiment using the test image `Building` from [7], which contains large rotational blurring that can be challenging for blind deblurring algorithms. Figure 3 (*middle row*) reveals that our algorithm contains less ringing and more fine details relative to Gupta *et al.*

**Comparison with Joshi *et al.* [13]:** Joshi *et al.* presents a deblurring algorithm that relies upon additional hardware for estimating camera motion [13]. However, even without this additional in-

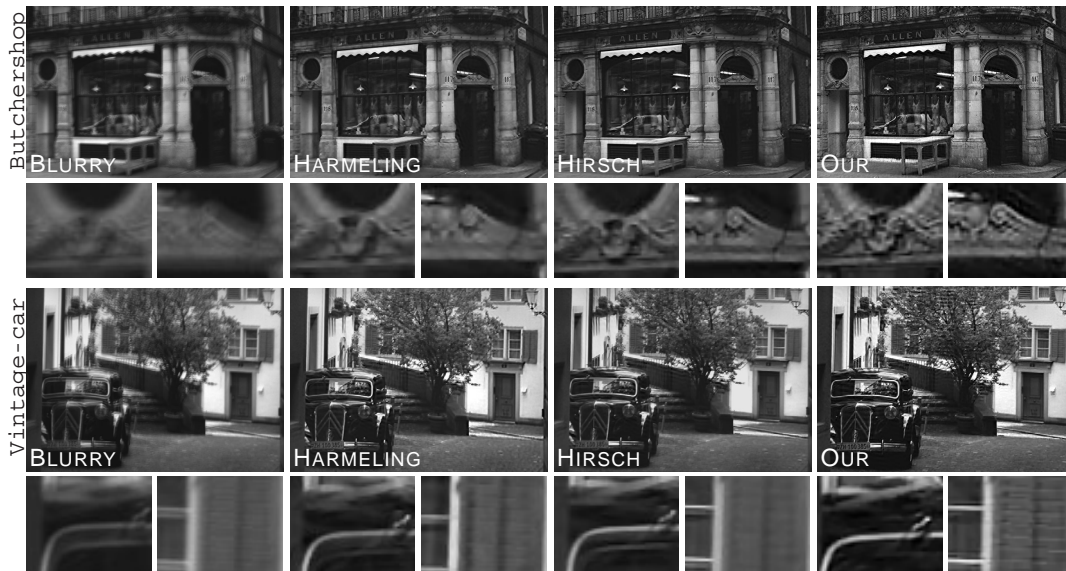

Figure 2: **Non-uniform deblurring results.** Comparison with Harmeling [8] and Hirsch [9] on real-world images. (better viewed electronically with zooming)

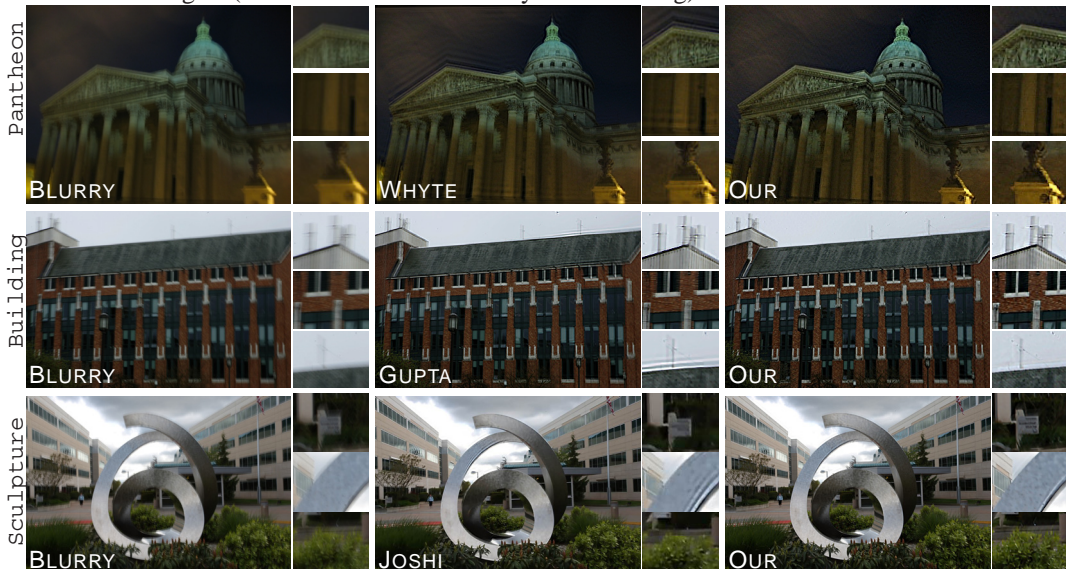

Figure 3: **Non-uniform deblurring results.** Comparison with Whyte [25], Gupta [7], and Joshi [13] on real-world images. (better viewed electronically with zooming)

formation, our algorithm produces a better sharp estimate of the Sculpture image from [13], with fewer ringing artifacts and higher resolution details. See Figure 3 (*bottom row*).

## 6   Conclusion

This paper presents a strikingly simple yet effective method for non-uniform camera shake removal based upon a principled, transparent cost function that is open to analysis and further extensions/refinements. For example, it can be combined with the model from [29] to perform joint multi-image alignment, denoising, and deblurring. Both theoretical and empirical evidence are provided demonstrating the efficacy of the blur-dependent, spatially-adaptive sparse regularization which emerges from our model. The framework also suggests exploring other related cost functions that, while deviating from the original probabilistic script, nonetheless share similar properties. One such simple example is a penalty of the form $\sum_i \log(\sqrt{\lambda} + |x_i| \|\bar{\mathbf{w}}_i\|_2)$; many others are possible.

**Acknowledgements**

This work was supported in part by National Natural Science Foundation of China (61231016).

## Footnotes

[1]The derivative filters used in this work are $\{[-1, 1], [-1, 1]^T\}$. Other choices are also possible.

[2] Note that even if the true sharp image is not exactly sparse, as long as it can be reasonably well-approximated by some exactly sparse image in an $\ell_2$ norm sense, then the analysis here still holds [27].

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
