[Supplementary Material]

# Non-Uniform Camera Shake Removal Using a Spatially-Adaptive Sparse Penalty Supplementary Material

**Haichao Zhang**[†‡] **and David Wipf** [§]

[†] School of Computer Science, Northwestern Polytechnical University, Xi'an, China
[‡] Department of Electrical and Computer Engineering, Duke University
[§] Visual Computing Group, Microsoft Research Asia, Beijing, China
`hczhang1@gmail.com`   `davidwipf@gmail.com`

## Abstract

This supplementary file provides the derivation of the proposed algorithm for non-uniform camera shake removal, as well as some implementation details, the proof of Theorem 1, and more extensive experimental comparisons.

## 1   Algorithm Derivation

Blind deblurring is achieved by minimizing the cost function from (7) or equivalently (8) in the main text. This can be accomplished by minimizing a rigorous upper bound $\mathcal{L}(\mathbf{x}, \mathbf{w}, \boldsymbol{\gamma})$ defined as

$$\mathcal{L}(\mathbf{x}, \mathbf{w}, \boldsymbol{\gamma}, \lambda) \triangleq \frac{1}{\lambda}\|\mathbf{y} - \mathbf{H}\mathbf{x}\|_2^2 + \sum_i \left[ \frac{x_i^2}{\gamma_i} + \log(\lambda + \gamma_i\|\bar{\mathbf{w}}_i\|_2^2) \right] + (n - m)\log\lambda, \tag{S1}$$

which is obtained by using the fact that

$$\psi(z, \lambda) = \min_{\gamma_i \geq 0} \frac{z^2}{\gamma} + \log(\lambda + \gamma_i) \tag{S2}$$

along with some additional algebraic manipulations. This expression can be shown by optimizing over $\gamma_i$, plugging in the resulting value which can be obtained in closed-form, and then simplifying. $\mathcal{L}(\mathbf{x}, \mathbf{w}, \boldsymbol{\gamma}, \lambda)$ can be iteratively minimized by optimizing $\mathbf{x}$, $\mathbf{w}$, $\boldsymbol{\gamma}$, and $\lambda$ with similar convergence properties to the EM algorithm. We now detail each constituent subproblem which are iterated until convergence.

**x-subproblem:** With other variables fixed, the latent image $\mathbf{x}$ is estimated via weighted least squares giving

$$\mathbf{x}^{\mathrm{opt}} = \left[ \frac{\mathbf{H}^T\mathbf{H}}{\lambda} + \boldsymbol{\Gamma}^{-1} \right]^{-1} \frac{\mathbf{H}^T\mathbf{y}}{\lambda}, \tag{S3}$$

where $\boldsymbol{\Gamma} = \mathrm{diag}[\boldsymbol{\gamma}]$. This can be computed efficiently using EFF and fast Fourier transforms [6].

**$\boldsymbol{\gamma}$-subproblem:** The optimization over each $\gamma_i$ is separable, thus can be solved independently via

$$\min_{\gamma_i \geq 0} \left[ \frac{x_i^2}{\gamma_i} + \log\left(\lambda + \gamma_i\|\bar{\mathbf{w}}_i\|_2^2\right) \right]. \tag{S4}$$

We can rewrite (S4) equivalently as

$$\min_{\gamma_i \geq 0} \left[ \frac{x_i^2}{\gamma_i} + \log\gamma_i + \log\left( \frac{\|\bar{\mathbf{w}}_i\|_2^2}{\lambda} + \gamma_i^{-1} \right) \right], \tag{S5}$$

where the irrelevant $\log \lambda$ term has been omitted. As no closed form solution is available for (S5), we instead use principles from convex analysis to form the strict upper bound

$$\frac{z_i}{\gamma_i} - \phi^*(z_i) \geq \log \left( \frac{\|\bar{\mathbf{w}}_i\|_2^2}{\lambda} + \gamma_i^{-1} \right), \quad \forall z_i \geq 0, \tag{S6}$$

where $\phi^*(z_i)$ is the concave conjugate of the concave function $\phi(\alpha) \triangleq \log(\frac{\|\bar{\mathbf{w}}_i\|_2^2}{\lambda} + \alpha)$. It can be shown that equality in (S6) is achieved when

$$z_i^{\mathrm{opt}} = \left. \frac{\partial \phi}{\partial \alpha} \right|_{\alpha = \gamma_i^{-1}} = \frac{1}{\frac{\|\bar{\mathbf{w}}_i\|_2^2}{\lambda} + \gamma_i^{-1}}, \quad \forall i. \tag{S7}$$

Substituting (S6) into (S5), we obtain the revised subproblem

$$\min_{\gamma_i \geq 0} \left[ \frac{x_i^2 + z_i}{\gamma_i} + \log \gamma_i \right], \tag{S8}$$

which admits the closed-form optimal solution

$$\gamma_i^{\mathrm{opt}} = {x_i}^2 + z_i. \tag{S9}$$

**w-subproblem:** Isolating $\mathbf{w}$-dependent terms produces the quadratic minimization problem

$$\min_{\mathbf{w} \geq 0} \frac{1}{\lambda} \|\mathbf{y} - \mathbf{Dw}\|_2^2 + \sum_i \log \left( \frac{\|\bar{\mathbf{w}}_i\|_2^2}{\lambda} + \gamma_i^{-1} \right). \tag{S10}$$

Because again there is no closed-form solution, we resort to similar bounding techniques as used above, incorporating the bound

$$\|\bar{\mathbf{w}}_i\|_2^2 v_i - \psi^*(v_i) \geq \log \left( \frac{\|\bar{\mathbf{w}}_i\|_2^2}{\lambda} + \gamma_i^{-1} \right), \quad \forall v_i \geq 0, \tag{S11}$$

where $\psi^*$ is the concave conjugate of the concave function $\psi(\alpha) \triangleq \log(\frac{\alpha}{\lambda} + \gamma_i^{-1})$. Similar to before, equality is achieved with

$$v_i^{\mathrm{opt}} = \left. \frac{\partial \psi_i}{\partial \alpha} \right|_{\alpha = \|\bar{\mathbf{w}}_i\|_2^2} = \frac{z_i}{\lambda}, \quad \forall i. \tag{S12}$$

Plugging (S11) into (S10), we obtain the minimization problem

$$\begin{aligned}
& \min_{\mathbf{w} \geq 0} \frac{1}{\lambda} \|\mathbf{y} - \mathbf{Dw}\|_2^2 + \sum_i v_i \|\bar{\mathbf{w}}_i\|_2^2 \\
& = \min_{\mathbf{w} \geq 0} \|\mathbf{y} - \mathbf{Dw}\|_2^2 + \mathbf{w}^T \left( \sum_i z_i \mathbf{B}_i^T \mathbf{B}_i \right) \mathbf{w}
\end{aligned} \tag{S13}$$

which can be solved efficiently using standard convex programming techniques.

$\lambda$**-subproblem:** Finally, the update rule for the noise level $\lambda$ can be obtained through similar analysis. Omitting the terms irrelevant to $\lambda$ we must solve

$$\min_{\lambda \geq 0} \frac{1}{\lambda} \left( \|\mathbf{y} - \mathbf{Hx}\|_2^2 + d \right) + n \log \lambda + \sum_i \log \left( \frac{\|\bar{\mathbf{w}}_i\|_2^2}{\lambda} + \gamma_i^{-1} \right), \tag{S14}$$

where $n$ is the dimensionality of $\mathbf{y}$ and we have added a small constant $d$ to the quadratic data fit term to prevent it from ever going to exactly zero. As before there is no closed-form solution, so we invoke the bound

$$\frac{\beta}{\lambda} - \varphi^*(\beta) \geq \sum_i \log \left( \frac{\|\bar{\mathbf{w}}_i\|_2^2}{\lambda} + \gamma_i^{-1} \right), \quad \forall \beta \geq 0, \tag{S15}$$

where $\varphi^*$ is the concave conjugate of $\varphi(\alpha) \triangleq \sum_i \log \left( \alpha \|\bar{\mathbf{w}}_i\|_2^2 + \gamma_i^{-1} \right)$, Equality is achieved with

$$\beta^{\mathrm{opt}} = \left. \frac{\partial \varphi}{\partial \beta} \right|_{\beta = \lambda^{-1}} = \sum_i \frac{\|\bar{\mathbf{w}}_i\|_2^2}{\frac{\|\bar{\mathbf{w}}_i\|_2^2}{\lambda} + \gamma_i^{-1}}. \tag{S16}$$

Plugging (S15) into (S14), we obtain the problem

$$\min_{\lambda \geq 0} \frac{1}{\lambda} \left( \|\mathbf{y} - \mathbf{Hx}\|_2^2 + d \right) + n \log \lambda + \frac{\beta}{\lambda} - \phi^*(\beta), \tag{S17}$$

leading to the closed-form noise level update

$$\lambda^{\mathrm{opt}} = \frac{\|\mathbf{y} - \mathbf{Hx}\|_2^2 + \beta + d}{n}. \tag{S18}$$

Note that $\lambda^{\mathrm{opt}}$ has a lower bound of $d/n$. Thus we may set $d$ so as to reflect some expectation regarding the minimum possible amount of noise or modeling error. In practice we simple choose $d = n10^{-4}$ for all experiments. More details regarding the issues involved in learning the noise can be found in [18].

## 2    Implementation Details

For practical simplicity, we only use projection operators $\mathbf{P}_j$ involving in-plane translations and rotations similar to [5] for modeling the camera shake, and use the EFF model [6] for reducing the computational expense. We have also incorporated a technique similar to the one used in [8], whereby irrelevant projection operators are pruned out while some new ones are added by sampling around the remaining projections using a Gaussian distribution with a small variance. Note that this heuristic is only for reducing the computational complexity; using the fully sampled basis set would generate the best results. Also, a standard multi-scale estimation scheme is incorporated consistent with most recent blind deblurring work [4, 12, 16, 7]. Finally, we emphasize that after the blur parameters are estimated using gradient images, a final non-blind deconvolution step is needed to recover the latent sharp image. We used the algorithm from [11] modified for the non-uniform domain.

## 3    Proof of Theorem 1

Recall that

$$\psi(u, \lambda) \triangleq \frac{2u}{u + \sqrt{4\lambda + u^2}} + \log\left(2\lambda + u^2 + u\sqrt{4\lambda + u^2}\right) \quad u \geq 0. \tag{S19}$$

First we want to show that $\psi(u, \lambda_1) \prec \psi(u, \lambda_2)$ given $\lambda_1 < \lambda_2$. For this purpose it is sufficient to demonstrate that $\frac{\partial^2 \psi(u,\lambda)}{\partial u^2} / \frac{\partial \psi(u,\lambda)}{\partial u}$ is an increasing function of $\lambda$, which represents an equivalent condition for relative concavity to one given by Definition 1 [14].

To this end, it is useful to first re-express $\psi(u, \lambda)$ using the equivalent variational form

$$\psi(u, \lambda) = \min_{\gamma \geq 0} \frac{u^2}{\gamma} + \log(\lambda + \gamma), \quad \forall u \geq 0, \tag{S20}$$

which can be verified straightforwardly by calculating the minimizing $\gamma^{\mathrm{opt}}$ and plugging it back into (S20). We further define

$$k^\lambda(v) \quad \triangleq \quad \psi(\sqrt{u}, \lambda) = \min_{\gamma \geq 0} \frac{v}{\gamma} + \log(\lambda + \gamma). \tag{S21}$$

Using results from convex analysis and conjugate duality, it can be shown that the minimizing $(\gamma_\lambda^{\mathrm{opt}})^{-1}$ for (S21) represents the gradient of $k^\lambda(v)$ with respect to $v$, meaning $\frac{\partial k^\lambda(v)}{\partial v} \equiv (\gamma_\lambda^{\mathrm{opt}})^{-1}$. Then we can compute the explicit expression for $\frac{\partial \psi(u,\lambda)}{\partial u}$ as

$$\frac{\partial \psi(u, \lambda)}{\partial u} = 2u \left.\frac{\partial k^\lambda(v)}{\partial v}\right|_{v=u^2} = \frac{u}{\lambda}\left(\sqrt{1 + \frac{4\lambda}{u^2}} - 1\right). \tag{S22}$$

Using (S22) it is also straightforward to derive $\frac{\partial^2 \psi(u,\lambda)}{\partial u^2}$ as

$$\frac{\partial^2 \psi(u, \lambda)}{\partial u^2} = 2\left.\frac{\partial k^\lambda(v)}{\partial v}\right|_{v=u^2} - \frac{4}{u^2\sqrt{1 + \frac{4\lambda}{u^2}}}. \tag{S23}$$

We must then show that

$$\frac{\partial^2 \psi(u,\lambda)/\partial u^2}{\partial \psi(u,\lambda)/\partial u} = \frac{1}{u} - \frac{\frac{4}{u^2\sqrt{1+\frac{4\lambda}{u^2}}}}{\frac{u}{\lambda}\left(\sqrt{1+\frac{4\lambda}{u^2}}-1\right)} \tag{S24}$$

is an increasing function of $\lambda$. By neglecting irrelevant additive and multiplicative factors (and recalling that $u \geq 0$), this is equivalent to showing that

$$\xi(\lambda) = \frac{1}{\lambda}\left(\sqrt{1+\frac{4\lambda}{u^2}}-1\right) \tag{S25}$$

is a decreasing function of $\lambda$. It is easy to check that

$$\xi'(\lambda) = \frac{\sqrt{1+\frac{4\lambda}{u^2}}-1-\frac{2\lambda}{u^2}}{\sqrt{1+\frac{4\lambda}{u^2}}} < 0. \tag{S26}$$

Therefore, $\xi(\lambda)$ is a decreasing function of $\lambda$, implying that $\frac{\partial^2\psi(u,\lambda)}{\partial u^2}/\frac{\partial\psi(u,\lambda)}{\partial u}$ is an increasing function of $\lambda$, thus $\psi(u,\lambda_1) \prec \psi(u,\lambda_2)$ for $\lambda_1 < \lambda_2$.

For the second property, when $\lambda$ becomes large, from (S22),

$$\frac{\partial\psi(z,\lambda)}{\partial z} = \frac{z}{\lambda}\left(\sqrt{1+\frac{4\lambda}{z^2}}-1\right) \rightarrow \frac{2}{\sqrt{\lambda}}, \quad \forall z \geq 0. \tag{S27}$$

Therefore, if the gradient of $\psi(z,\lambda)$ with respect to $z \geq 0$ is constant (and symmetrically $-z$), then $\sum_i \psi(|z_i|,\lambda) \rightarrow \frac{2\|\mathbf{z}\|_1}{\sqrt{\lambda}}$. Perhaps more importantly, when we multiply both sides of (8) in the main text by $\lambda$, then the effective image penalty scales as $2\sqrt{\lambda}\|\mathbf{z}\|_1$ leading to (10) in the main text.

On the contrary, when $\lambda \rightarrow 0$, both terms in (S19) converge to $\ell_0$ norm of $\mathbf{z}$ when summed over $i$, ignoring some irrelevant constant factors, as can be inferred from [17], completing the proof. ∎

## 4   Additional Experimental Results

This section provides more deblurring results involving both uniformly and non-uniformly blurred images.

### 4.1   Uniform Deblurring

Any non-uniform deblurring approach should naturally reduce to an effective uniform algorithm when the blur transformations are simply in-plane translations. We first evaluate the proposed algorithm in the uniform case where existing benchmarks facilitate quantitative comparisons with state-of-the-art methods. For this purpose we reproduce the experiments from [12] using the benchmark test data from [13],[1] which consists of 4 base images of size $255 \times 255$ and 8 different blurring effects, leading to a total of 32 blurry images. Ground truth blur kernels were estimated by recording the trace of focal reference points on the boundaries of the sharp images. The kernel sizes range from $13 \times 13$ to $27 \times 27$. We compare the proposed method with only in-plane translation, with the algorithms of Shan *et al.* [15], Xu *et al.* [19], Cho *et al.* [3], Fergus *et al.* [4], Levin *et al.* [12] and Babacan *et al.* [1]. The SSD (Sum of Squared Difference) metric defined in [13] is used for measuring the error between estimated and the ground-truth images. To normalize for the fact that a harder kernel gives a larger image reconstruction error even when the true kernel is known (because the corresponding non-blind deconvolution problem is also harder), the SSD ratio between the image deconvolved with the estimated kernel and the image deconvolved with the ground-truth kernel is used as the final evaluation measure. The cumulative histogram of the error ratios is shown in Figure 1. The height of the bar indicates the percentage of images having error ratio below that level. Higher bars indicate better performance, revealing that the proposed method significantly outperforms existing methods on uniform deblurring tasks.

Figure 1: Evaluation of uniform deblurring results using cumulative histogram of the deconvolution error ratio across 32 test examples from [13]. The height of the bar indicates the percentage of images having error ratio below that level. Higher bars indicate better performance.

## 4.2   Non-Uniform Deblurring

For non-uniform deblurring, quantitative comparisons are much more difficult because of limited benchmark data with available ground truth.[2] Moreover, because source code for most state-of-the-art non-uniform algorithms is not available, it is not feasible to even qualitatively compare all methods across a wide range of images. Consequently, a feasible alternative is simply to visually compare our algorithm using images contained in previously published papers with the deblurring results presented in those papers.

Here we provide non-uniform deblurring results on real-world images as well as kernel comparisons where ground-truth is available (see Figure 2 and Figure 3). Although some images already appeared in the main text, they are reproduced here for ease of comparison. We also compare with another recent non-uniform deblurring algorithm and images from Cho *et al.* [2] which were not included in the main text due to space limitations (Figure 4-Figure 7). Given all of the examples below, the proposed method generally performs comparably or better than existing methods, despite being virtually parameter free and having arguably the simplest formulation.

## Footnotes

[1] http://www.wisdom.weizmann.ac.il/~levina/papers/LevinEtalCVPR09Data.rar

[2]There is a recent dataset from Kohler *et al.* [10] that contains non-uniformly blurred images. However, because a uniform deblurring algorithm presently performs best on this data we decided not to pursue evaluation further. Note that our paper explicitly intends to address the scenario where uniform deblurring algorithms are demonstrably inadequate.

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

Figure 3: Non-uniform kernel estimation comparisons associated with the `Butchershop`, `Vintage-car`, and `Elephant` images from Figure 2 above. Estimated blur kernels are resized for display purposes. The ground-truth local blur kernel patterns for these three test cases were measured by creating a color image where the gray scale image is shown in the red channel, a delta peak image for recording the blur kernels is shown in the blue channel, and the green channel is set to zero. This color image is then displayed on a computer screen for capturing using a hand-held camera. (See [6] for more details on this process.) The estimated kernel patterns are shown using $7 \times 9$ (or $9 \times 7$) arrays for visualization. Inspection of these patterns reveals that the estimates from our algorithm resemble the ground-truth more reliably, a further indication of its effectiveness.

Figure 4: Non-uniform deblurring comparisons with Whyte [16] and Hirsch [7] on the real-world images `Pantheon` and `Statue` from [16]. On the `Pantheon` example the deblurring result from Whyte *et al.* has significant ringing artifacts while the result from Hirsch *et al.* seems to be suffering from some chrome distortions as indicated by the dome area of the pantheon. Our result on the other hand, has very few artifacts or chrome distortions. On the `Statue` image the result of Whyte *et al.* is generated using a blurry image paired with another additional noisy image of the same scene captured with a shorter exposure time length. Our method and that of Hirsch *et al.*, without the benefit of such additional image data, can nonetheless generate a deblurring result with comparable quality.

Figure 5: Non-uniform deblurring comparisons with Gupta [5] and Hirsch [7] on the real-world images Magazines and Building from [5]. Note that Hirsch *et al.* do not provide a deblurring result for the Building image. Overall, the proposed method performs comparably or better than the other methods.

Figure 6: Non-uniform deblurring comparisons with Joshi [9] and Harmeling [6] using the real-world images Porsche and Sculpture provided by Joshi *et al.* [9]. Note that Harmeling *et al.* do not provide a deblurring result for the Sculpture image. Our approach generally recovers more detail with less ringing than the other methods, despite the fact that Joshi *et al.* use additional hardware to estimate camera motion.

Figure 7: Non-uniform deblurring comparisons with Cho *et al.* [2] on the real-world images Antefix and Doll from [2]. Note that the method of Cho *et al.* requires two blurry images as input while we ran our algorithm using only the first blurry image in each test pair. Despite this significant disadvantage, our method still produces higher quality estimates in both cases.