[Reviews · NeurIPS 2013]

Submitted by Assigned_Reviewer_4

In this paper, the authors develop a novel blind deconvolution algorithm that provides very good results for both uniform and non-uniform blind deconvolution problems. They provide a compelling analysis of the reasons why their method works well. A number of experimental results are provided against the state-of-the art algorithms for both uniform and non-uniform cases. The paper is clear and elegantly written. I believe this is a significant contribution to the literature on blind deconvolution.

It would be interesting if the authors discussed how their new technique relates to the variational deblurring algorithm of Levin et. al. [15], which also uses marginalization over x to determine the blur kernel. In fact, the authors' new formulation starts with an even more simplified prior than the heavy tailed one used by Levin et. al. However, instead of fixed variances as used in Levin et. al., the authors make these variances spatially varying hyper-parameters...is this the crucial difference?
Summary: A novel and interesting new algorithm for blind deconvolution is presented.

Submitted by Assigned_Reviewer_5

This paper describes a method for non-uniform (space-varying) blind deconvolution.
The method is strongly based on ref [26]. Indeed the authors propose to solve
eq. 9, which is essentially the same as in eq. 26 and 27 in ref [26], by using the
upper bound derived in eq 7, which is one of the novel bits.
The analysis of the effects of such prior, however, is carried out on the original cost.
A summary of the analysis is that the proposed prior adaptively regulates
the concavity/convexity of the image-blur prior depending on the
magnitude of the local image gradients and the L2 norm of the blur.
If the blur is given at any pixel, the theory is that a high L2 norm (up to 1) can only be achieved
with a Dirac delta and hence the image is sharp to start with. Here a very concave prior which
strongly induces gradient sparsity is welcome. Vice versa, where the blur tends to be an averaging
kernel, the prior tends to be much less concave to allow for less sensitivity to fine details and more
sensitivity to coarse details.

Quality
--
The algorithm and the underlining theory are interesting and compelling.
Although a good portion of the paper is devoted to explaining the effect of the
proposed prior, there are several points left unexamined.
1 - The analysis is carried out on the original cost 9 but
the effects of the bound 7, which is the novelty in the algorithm, are not discussed.
2- It would be useful to see how the analysis changes under the uniform case (that would
complement well with the experiments in Fig 2).
3- In light of the results in ref [16] it would be useful to see
a discussion of how this prior addresses the limitations of the classic
priors; specifically, it would be interesting to see that the blurry image & no blur
solution is no longer a global minimum, or, even better, no longer a local minimum.


Clarity
--
Overall the authors do a good job with explaining the approach.
However, there is somehow a jump between the paragraphs in sec 3.
For example, the connection between eq 7 (the approx) and cost 9
is not made clear. Indeed eq. 7 is never used later on. One can
then find it only in the supplementary material.
I recommend to revise/rewrite this section.
The use of w for the weights and \hat w for the blur kernels is
quite confusing. Despite the relation in eq 8, their meaning is
very different. Please consider changing one of the two (e.g. h_i for \hat w_i
would be much more meaningful).
Eq 6 might have some typos: check that T at the exponent is not -1 and that
you are not missing a product over i under the integral.



Originality
--
The originality is limited by all the body of work by ref [26].
It probably would have been very useful to discuss the differences
with respect to [26]. In my opinion the originality is limited to a bound (eq7)
and an explanation of how this prior operates (via Theorem 1).
The explanation however, is quite approximate due to the
complexity of the prior.


Significance
--
The study and development of novel priors for blind deconvolution is quite important
and this paper further develops the new direction introduced by ref [26].
Moreover, given the experimental performance of this algorithm, this approach
deserves attention.



Summary: Overall this paper introduces some novel elements: a practical bound for a cost function that simplifies the implementation, and
analysis that explains the general behavior of the adaptive prior. The performance is quite good.

Submitted by Assigned_Reviewer_6

The paper addresses the problem of single image blind deconvolution with non-uniform blur caused by camera shake. The authors propose a two step procedure: first to estimate the motion blur and second the recovery of the sought-after latent image through non-blind deconvolution. This procedure is common practice in blind deconvolution. The main contribution made here is the derivation of a Bayesian inference strategy for motion blur estimation. In a number of experiments the authors demonstrate the validity of their approach and compare its performance against other state-of-the-art methods. The mathematical derivation appears sound and the proposed scheme is claimed to exhibit a number of favourable properties. Although these details are discussed in some detail in the second half of Section 4, no empirical evidence is given, which would strengthen the argumentation and verify the claims made. The authors also miss to make connections to other recent approaches that advocate a similar line of reasoning, in particular [15] and the missing reference:

S.D.Babacan, et al., "Bayesian Blind Deconvolution with General Sparse Image Priors", ECCV 2012

In addition, the authors miss another relevant reference, namely

R.Kohler, et al., "Recording and playback of camera shake: Benchmarking blind deconvolution with a real-world database." ECCV 2012.

that would allow them to benchmark their approach against other state-of-the-art single image blind deblurring methods for both uniform and non-uniform blur. Besides, a number of questions remain unanswered (see below) in the presented exposition, that lower the overall quality of the paper.

The paper is clearly structured and well written. Section 4 would benefit from a division in smaller subsections as it feels a bit lengthy for my tasting. Unfortunately, some important details are missing (e.g. which algorithm is used for the final non-blind deconvolution, see also comments below), which impair the overall clarity of the paper.

The theoretical foundation of the proposed approach build on previous work [19,26], however its application to the problem of single image blind deconvolution seems novel.

Although the presented algorithm seems to deliver comparable results to state-of-the-art algorithms, the provided insights are very limited. It remains unclear in which situations/cases the proposed method works better and why. It also fails to make connections to other recent relevant work (see above) and place its contribution into context, which lowers its significance considerably.

Further comments:
* What method is employed for the final non-blind deconvolution?
* It is stated, that the approach is "parameter-free". Is this also true for the kernel size?
* It is mentioned in Section 4, that initialization with a large $\lambda$ and its subsequent evolution render the deconvolution process more stable. It would be interesting to see a plot that shows the evolution of $\lambda$ in the course of the motion blur estimation process.
* What basis (i.e. $B$ in Eq.(8)) is chosen for the experiments with non-uniform blur? Is $w^\bar_i$ evaluated for every pixel? If not, how are the evaluation locations chosen?
* From the main paper it is not clear that a coarse-to-fine scheme is employed. Great if this could be made clearer.
* What are typical run-times for the proposed method?
* Large blur is challenging even for state-of-the-art methods. How does performance scale with kernels size?
* The ringing artifacts in Whyte's result in Fig.4 are stemming primarily from the final non-blind deconvolution (see his talk slides). It would be more fair to use the one published on the accompanying project webpage (http://www.di.ens.fr/willow/research/deblurring/).
* The correct reference for the efficient filter flow framework mentioned in Section 2 is
M.Hirsch, et al., "Efficient filter flow for space-variant multiframe blind deconvolution." CVPR 2010.
* What are the limitations of the proposed approach?
Summary: The paper presents an interesting Bayesian approach to motion blur estimation. Unfortunately, it fails to make connections to relevant previous work and lacks a quantitative evaluation for the case of camera shake removal with non-uniform blur, which lower its significance and value to the community considerably.
Author Feedback

Author rebuttal: The revised version of our paper and the associated supplementary material address the major concerns of the reviewers and clarify other points and analyses that were previously unclear.